# Recent Advances in the Molecular Biology of Chronic Lymphocytic Leukemia: How to Define Prognosis and Guide Treatment

**DOI:** 10.3390/cancers16203483

**Published:** 2024-10-14

**Authors:** Annalisa Arcari, Lucia Morello, Elena Borotti, Elena Ronda, Angela Rossi, Daniele Vallisa

**Affiliations:** 1Hematology Unit, Ospedale Guglielmo da Saliceto, Azienda USL di Piacenza, 29100 Piacenza, Italy; l.morello@ausl.pc.it (L.M.); d.vallisa@ausl.pc.it (D.V.); 2Bone Marrow Transplant Laboratory, Molecular Diagnostic and Stem Cells Manipulation, Ospedale Guglielmo da Saliceto, Azienda USL di Piacenza, 29100 Piacenza, Italy; e.borotti@ausl.pc.it (E.B.); e.ronda@ausl.pc.it (E.R.); a.rossi3@ausl.pc.it (A.R.)

**Keywords:** chronic lymphocytic leukemia, precision medicine, minimal residual disease

## Abstract

**Simple Summary:**

The therapeutic landscape of Chronic Lymphocytic Leukemia (CLL) has dramatically changed in recent years, with a shift from chemoimmunotherapy towards many new targeted agents, such as Bruton Tyrosine Kinase inhibitors (BTKi) and anti-BCL-2. We now need an accurate re-definition of prognostic/predictive parameters to support clinicians in choosing the more appropriate option. This review explores recent advances in the molecular biology of chronic lymphocytic leukemia, with the aim of identifying the most important biomarkers to define prognosis and to guide first-line and second-line treatment.

**Abstract:**

Chronic Lymphocytic Leukemia (CLL) is the most frequent type of leukemia in Western countries. In recent years, there have been important advances in the knowledge of molecular alterations that underlie the disease’s pathogenesis. Very heterogeneous prognostic subgroups have been identified by the mutational status of immunoglobulin heavy variable genes (*IGVH*), FISH analysis and molecular evaluation of *TP53* mutations. Next-generation sequencing (NGS) technologies have provided a deeper characterization of the genomic and epigenomic landscape of CLL. New therapeutic targets have led to a progressive reduction of traditional chemoimmunotherapy in favor of specific biological agents. Furthermore, in the latest clinical trials, the minimal residual disease (MRD) has emerged as a potent marker of outcome and a guide to treatment duration. This review focuses on recent insights into the understanding of CLL biology. We also consider the translation of these findings into the development of risk-adapted and targeted therapeutic approaches.

## 1. Introduction

Chronic lymphocytic leukemia (CLL) is the most frequent lymphoproliferative disorder in Western countries, accounting for 1% of all cancer cases [1]. The estimated number of new cases in 2024 in the United States is 20,700, with 4440 estimated deaths. The median age at diagnosis is 70 years, with most new cases among people aged 65–74 (32.3%) and 75–84 (24.6%) [1]. Outcomes have continuously improved during the past decades [2,3], with a 5-year relative survival of 88% in 2014–2020 [1].

The 5th edition of the World Health Organization (WHO) classification of hematolymphoid tumors and the International Consensus Classification (ICC) of mature lymphoid neoplasms agree in defining CLL as a low-grade lymphoproliferative neoplasm with ≥5 × 10^9^/L clonal B-cells in the peripheral circulation that express CD5, CD19, CD20(dim), and CD23 [4,5]. The diagnosis is based on morphological and flow cytometric analysis of peripheral blood. A frequent pre-neoplastic condition named Monoclonal B-cell Lymphocytosis (MBL) can be detected when these clonal B-cells are below 5 × 10^9^ in the absence of lymphadenopathy, organomegaly, and cytopenias [6].

CLL has a very heterogeneous clinical behavior. Most patients experience an indolent form of the disease, characterized by slow progression and minimal symptoms, allowing them to live for many years without treatment. Conversely, about one-third of patients present a more rapid disease progression and significant symptoms requiring an early therapeutic intervention. From a biological point of view, this variability is influenced by multiple factors, including structural genomic aberrations, molecular mutations, epigenetic changes, and microenvironment interactions. Understanding this heterogeneity is crucial for developing personalized treatment strategies and improving patient outcomes.

So far, early intervention has not demonstrated a beneficial impact on the survival of asymptomatic CLL patients [7]. The current standard for early-stage patients remains a watch-and-wait approach even with an unfavorable genetic risk profile; only those patients who meet the 2018 International Workshop on Chronic Lymphocytic Leukemia (iwCLL) criteria for initiation of therapy should receive treatment [8].

The therapeutic landscape of CLL has significantly evolved in recent years. The use of immunochemotherapy is currently very limited, given the advent of many new biological agents. These novel targeted agents, including Bruton Tyrosine Kinase (BTKi) inhibitors (e.g., ibrutinib, acalabrutinib, and zanubrutinib) and B-cell Lymphoma-2 (BCL-2) inhibitors (e.g., venetoclax) have shown substantial efficacy and improved safety profiles compared to traditional immunochemotherapy. This shift towards targeted treatments has transformed the management of CLL and needs an accurate re-definition of prognostic/predictive parameters. Furthermore, in the latest clinical trials, the minimal residual disease (MRD) has emerged as a potent marker of outcome and a tool to guide treatment duration. 

## 2. How to Define Prognosis

In recent years, several studies have led to a greater understanding of the genetic landscape of CLL, confirming that it is a heterogeneous and dynamic disease not associated with a single genetic lesion but with several molecular abnormalities [9,10].

Combining different clinical features with the most frequent genetic lesions, several prognostic scores have been developed to better stratify patients across different risk categories. The international prognostic index for chronic lymphocytic leukemia (CLL-IPI) combines five independent prognostic factors (*TP53* status, *IGVH* mutational status, serum beta2-microglobulin, clinical stage according to Rai/Binet, and age) identifying four risk groups with significantly different 5-year overall survival (OS), from 93% (low risk) to 23% (very high risk) [11]. A recent study has confirmed the prognostic value of CLL-IPI in predicting progression-free survival (PFS) but not OS in the era of targeted therapies [12]. Targeted therapies showed better outcomes compared to immunochemotherapy across various risk groups.

To date, according to the 2018 iwCLL guidelines [13], the following biomarkers should be tested before treatment start: (i) *TP53* aberrations, including both *TP53* mutation and 17p deletion, (ii) 11q deletion, 13q deletion and trisomy 12, and (iii) *IGHV* somatic hypermutation (SHM) status [8].

### 2.1. TP53 Aberrations 

*TP53* aberrations have a strong prognostic/predictive role in CLL patients, conferring a worse prognosis with all existing treatments. Interestingly, the effect of *TP53* status on treatment outcomes seems to be different with targeted agents compared to chemoimmunotherapy (see paragraph “How to guide treatment choice in first-line therapy”).

*TP53* aberrations include both the deletion of the *TP53* gene locus on 17p13 [del(17p)] and the presence of somatic variants of the *TP53* gene. Del(17p) is routinely detected by fluorescence in situ hybridization (FISH), while *TP53* mutations can be revealed through Sanger sequencing or next-generation sequencing (NGS) techniques. In CLL patients, del(17p) is mostly observed together with *TP53* mutations, while an isolated del(17p) is rarer; conversely, *TP53* mutations can also be found without del(17p) [14]. Therefore, both tests should be performed before each line of therapy, as *TP53* aberrations may appear during the disease course and after previous treatments. The European Research Initiative on CLL (ERIC) has implemented the *TP53* Network program with the aim of standardizing the laboratories that perform these tests in routine clinical practice [14]. Approximately 75% of mutations in *TP53* are single nucleotide variants (SNV), mainly in the DNA-binding domain of the p53 protein. Otherwise, mutations can also be observed in the oligomerization domain or C-terminal domain. Patients carrying missense mutations in the DNA binding domain show a shorter survival compared to patients with mutations outside that domain [15,16]. A smaller percentage of *TP53* mutations include frameshift, indels, nonsense and splice-site mutations. Although some *TP53* mutations produce a loss-of-function effect on the encoded protein, alternatively or in parallel, *TP53* missense mutations may also lead to a gain-of-function phenotype [14].

According to the ERIC guidelines, the mutational analysis of *TP53* should be performed following the 10% cut-off of variant allele frequency (VAF) [14]. However, the advent of NGS in routine practice allows the identification of low-burden *TP53* mutations (VAF < 10%). The current conventional threshold of 10% is arbitrary, and reporting variants with VAF up to 5% is currently allowed by ERIC, although still under evaluation. The clinical significance of low-burden *TP53* mutations is debated. Some studies suggest that patients with low VAF *TP53* clones have the same unfavorable clinical prognosis as patients with higher VAF [17,18]. The mutation burden increases during relapse after chemo- and/or immunotherapy, but persisting or diminishing alterations are also detected, mainly among 1% VAF variants [14]. A recent study on a “real-world” cohort of CLL patients proposes that clonal expansion may contribute to the inferior survival of cases carrying low-level *TP53* mutations. A higher expansion of these subclones is particularly observed in patients relapsing after fludarabine–cyclophosphamide-rituximab (FCR) with an impact on OS [17,18,19].

Another important aspect is the presence in many CLL patients of multiple *TP53* variants. Bi-allelic *TP53* inactivation could explain two *TP53* variants but not a higher number of them. This high intratumor heterogeneity has been observed in several studies and confirmed by single-cell sequencing [20]. Most of these multiple *TP53* variants are truly pathogenic. The basis of this specific selection for multiple *TP53* variants during the CLL course is currently not well defined, despite recent data suggesting a more aggressive clinical phenotype [20]. Single-hit *TP53* mutations still show a not clear prognostic significance with targeted agents, while concomitant multi-hit *TP53* aberrations appear to be independently associated with worse outcomes [21,22]. It will be very important to understand the real incidence of multiple *TP53* sub-clonal mutations and their role as a driver of subsequent relapses.

### 2.2. IGVH Mutational Status

The SHM status of the rearranged *IGHV* gene is one of the main gold standards for accurate risk stratification in CLL, and its evaluation is recommended before treatment [8]. Following ERIC recommendations, there are two categories: ≥98% germline identity for unmutated *IGHV* genes (UM) and <98% germline identity for mutated *IGHV* genes (M) [13,14,23]. Several studies demonstrate that UM-CLL patients generally show an inferior prognosis if compared with M-CLL ones who can survive decades without treatment intervention [14,15]. Particular attention is needed for 5% of cases showing homology with the germline sequence between 97% and 97.99%, defined as borderline *IGHV* (BL). The 98% cut-off is a purely mathematical convention rather than biological, even if it is useful to define CLL subgroups with distinct outcomes. To date, it is unclear if BL patients should be classified as M-CLL cases or not since the clinical outcome of this group remains poorly defined [24,25,26,27].

In the CLL prognosis definition, another important feature is the detection of different stereotyped subsets of the B-Cell Receptor (BCR). The most frequent subsets are named #1, #2, #8 (unfavorable prognosis), and #4 (good outcome). Subset #2 (IGHV3-21, IGLV3-21) may have important clinical implications, with dismal prognosis irrespective of *IGHV* mutational status. Subset #2 patients also have a higher incidence of *SF3B1* and *ATM* mutations/del(11q) [28].

Although the gold standard technique to assess *IGHV* mutational status is Sanger sequencing, NGS assays are widely used, enabling the identification of small sub-clones of cells in CLL populations. These sub-clones may also present discordant SHM status with different prognostic implications that are not clearly defined. Therefore, recent studies propose a revision of prognostic stratification with five categories: multi-M (multiple M-clones), M (single M-clone), mix M-UM (mix of M and UM), UM/VH3-21 (single UM-clone or presence of VH3-21 clonotype), and multi-UM (multiple UM-clones) [25,26,29].

### 2.3. Emerging Prognostic and Predictive Biomarkers 

Nowadays, whole-exome sequencing (WES) and whole-genome sequencing (WGS) studies allow the rapid identification of new biomarkers that drive the pathogenesis of CLL. Therefore, prognostic scores used in clinical practice should be integrated with new genetic-molecular parameters. More than 40 recurrently mutated driver genes have been identified in CLL. The involved pathways include microenvironment-dependent signaling through *NOTCH* (*NOTCH1*, *FBXW7*), inflammatory receptors (*MYD88*), MAPK–extracellular signal-regulated kinase (*BRAF*, *KRAS*, *NRAS*) and NF-κB pathways (*BIRC3*, *NFKBIE*), DNA damage and cell cycle control (*ATM*, *TP53*, *POT1*), and RNA processing (*XPO1*, *SF3B1*) [23,30,31,32] (Figure 1). The number of mutations impacts disease prognosis. More than two mutations are independently associated with a shorter Time to First Treatment (TTFT). Genetic dynamics in untreated CLL patients suggest that monitoring the VAF of a special gene panel may predict disease progression.

The emerging biomarkers are described below, according to the genetic pathway in which they are involved.

#### 2.3.1. Microenvironment-Dependent Signaling through NOTCH

##### *NOTCH1* 

*NOTCH1* variants occur in approximately 11% of untreated CLL patients, with increasing prevalence in disease progression and Richter transformation [33]. *NOTCH1* encodes a class I transmembrane protein acting as a ligand-activated transcription factor fundamental in cell proliferation, differentiation, and apoptosis. *NOTCH1* mediates transcriptional activation of multiple target genes, including *TP53*, *MYC* and components of the NF-kB pathway [33].

*NOTCH1* variants are mainly truncating or deletion mutations involving the C-PEST domain, sequence as coding mutations (p.P2514fs*4) or the 3′ UTR (c.*370A>G, c.*377A>G) as non-coding mutations. These variants lead to a constitutive activation of *NOTCH1* signaling [30,34]. In a study by the MD Anderson Cancer Center on 1574 CLL patients, *NOTCH1* mutations were detected in 252 cases. CLL patients with clonal coding and non-coding *NOTCH1* mutations had more frequently a UM-*IGHV* status and showed a significantly shorter TTFT than patients with wild type (wt)-*NOTCH1* [32]. *NOTCH1* may have an important role as a driver of CLL initiation and progression, becoming an independent poor prognostic factor for aggressive forms of disease [34,35,36]. In the German CLL8 trial, patients with *NOTCH1* mutations did not derive benefit from rituximab, probably due to a decreased surface expression of CD20 [35]. Since the key role of *NOTCH1* in CLL is that this gene provides an attractive target for therapeutic intervention, several *NOTCH1* inhibitors are in development.

##### *FBXW7* 

*FBXW7* encodes a negative regulator of *NOTCH1*. Approximately 5% of CLL patients show *FBXW7* mutations, which are mainly missense mutations located in the WD40 substrate binding domain. Their functional consequences are still not clearly defined: *FBXW7* mutations may result in an increase in *NOTCH1* target gene expression characterized by dysregulation of *NOTCH1* signaling. FBXW7 mutated patients seem to display poorer PFS and OS in patients with early-stage disease [37,38].

#### 2.3.2. RNA Processing

##### *SF3B1* 

*SF3B1* represents the largest subunit of the SF3B complex and functions as a core component of the U2 snRNP, crucial for the branch site recognition and for the first stages of assembly of the spliceosome. It is recurrently mutated in approximately 8% of CLL patients. *SF3B1* mutations mainly occur in the C-terminal HEAT-repeat domain (i.e., K700E); mutant SF3B1 seems to inhibit the canonical recognition by wild-type U1 snRNA on specific 5′ splice sites [39]. Several studies demonstrate that when combined with *ATM* deletion, *SF3B1* K700E is sufficient to cause CLL [40]. *SF3B1* alterations are associated with UM-*IGHV*, correlating to unfavorable prognosis with faster disease progression, poor OS and shorter TTFT. *SF3B1* are more frequently found as sub-clonal variants, and they are likely later events in CLL progression [21,36,41].

##### *XPO1* 

*XPO1* is a key mediator of nuclear export. *XPO1* mutations are identified in approximately 4% of CLL cases [42]. *XPO1* mutations cluster in two hot spots, probably disrupting highly conserved interactions in the nuclear export signal (NES) binding site, conferring novel cargo-binding abilities, and causing cellular mislocalization of critical regulators. *XPO1* mutations seem to present a dominant role in CLL initiation. Additional mutations are mainly found in *NOTCH1* or *SF3B1*, less in *TP53*; patients with *XPO1* mutations often carry also del(11q). Survival analysis suggests that *XPO1*-mutated patients have an inferior outcome. Moreover, *XPO1* mutations are independent prognostic variables in both U-CLL and M-CLL [31,39,42].

#### 2.3.3. DNA Damage and Cell Cycle Control

##### *ATM* 

*ATM* encodes a protein kinase, which functions as an upstream regulator of the *TP53* gene. The *ATM* gene is located on chromosome 11, and therefore, del(11q) commonly encompasses the *ATM* gene. Del(11q) is mainly monoallelic, and it is often associated with mutations in the remaining *ATM* allele. *ATM* mutations are reported in approximately 12% of CLL cases, and they may be a cooperative genetic event that occurs during tumor progression [43]. In CLL patients, somatic *ATM* mutations predict shorter TTFT and OS after chemoimmunotherapy [41,44]. Clonal evolution studies from the CLL8 trial suggest that the biallelic loss of *ATM* confers a poorer prognosis compared with monoallelic *ATM* [31]. Recent studies reveal a possible germline origin for some *ATM* mutations (L2307F) that could potentially influence clinical outcomes and contribute to the inherited risk for developing CLL; these data emphasize the importance of analyzing variants of unknown significance (VUS) and NGS constitutional data [45]. Interpretation of *ATM* gene mutation analysis is still challenging because of the large size of the gene, the high presence of VUS, and the possible detection of germline variants.

##### *POT1* 

*POT1* encodes a protein that protects chromosomal ends from an inappropriate DNA damage response. *POT1* mutations are identified in approximately 3% of CLL cases [46]. *POT1* somatic mutation generally affects key residues for telomeric DNA binding. Patients carrying *POT1* mutations are usually young and often have UM-*IGHV* with poor OS. The most frequently mutated genes co-occurring with *POT1* are *NOTCH1*, *TP53*, and *SF3B1*. A high proportion of *POT1*-mutated patients had a history of prior tumors, maybe suggesting a possible germline origin for these mutations [39,46]. Additional studies are needed to clarify their impact.

#### 2.3.4. NF-κB Signaling Pathways

##### *BIRC3* 

*BIRC3* encodes a negative regulator of the noncanonical NF-κB signaling pathway, showing a key role in CLL pathogenesis. *BIRC3* mutations are found in approximately 4% of CLL cases and are mostly truncating forms [36]. The role of *BIRC3* is not well defined, perhaps due to the low incidence of its mutations and their occurrence as a late event. Some studies show that, in unselected CLL patients, *BIRC3* mutations are associated with UM-*IGHV* and an unfavorable prognosis, while other studies do not support their prognostic significance [23,36,47]. *BIRC3* variants may also be evaluated as a predictive marker of response, associated with chemo-refractoriness to fludarabine-based combinations. Preliminary data suggest a better sensitivity to BCL-2 inhibitors [31,47].

##### *NFKBIE* 

*NFKBIE* encodes the IκBε, which is an inhibitor of NF-κB-inducible activity. Approximately 4–10% of untreated CLL patients reveal *NFKBIE* mutations, which typically are truncating mutations. How they contribute to CLL pathogenesis and progression is still not fully understood, and further studies are required [39,48]. Recent studies show that *NFKBIE* mutations could reduce the response to ibrutinib, often cooperating with *MYD88* mutations [49].

#### 2.3.5. Inflammatory Receptors

##### *MYD88* 

*MYD88* encodes an adapter protein that recruits interleukin-1 receptor-associated kinase 4 (IRAK4), constituting a complex that is a critical signaling mediator of the Toll-like receptor/interleukin-1 receptor superfamily. *MYD88* mutations are detected in approximately 2.5% of CLL patients [50]. *MYD88* mutations are strongly associated with young age at diagnosis, predominantly male, favorable cytogenetics, good prognosis, and M-*IGHV* status. CLL patients carrying *MYD88* mutations do not display a significant difference in TTFT compared to *MYD88* wt patients. *MYD88*-mutated cases rarely carry a mutation in a second gene [23,39,51].

#### 2.3.6. MAPK–Extracellular Signal-Regulated Kinase

##### *RAS/RAF* Genes

In CLL patients, variants in genes of the RAS-BRAF-MAPK-ERK pathway have not been fully studied. *RAF* mutations are observed in CLL patients with a frequency of approximately 3% [52]. The most frequent mutations seem to be missense, sub-clonal, and mutually exclusive and located within or near the activation loop. A recent study suggests that mutations in the RAS–BRAF–MAPK–ERK pathway are associated with adverse biological features such as UM-*IGHV* due to their upregulating action on the *RAS*-mediated signaling pathway. Significantly, the impact of *RAS* variants on TTFT was independent of *IGHV* status and the presence of variants in *TP53*, *ATM* or *BIRC3.* Finally, it was reported that *BRAF* mutations are related to adverse OS, while *KRAS* and *NRAS* alterations are not. The role of these genes in CLL pathogenesis and prognosis is yet under evaluation, and further studies are needed [52,53].

Prognostication in CLL is an active research field that defines not only the prognostic markers at diagnosis but also the predictive markers connected to treatment response in the era of targeted therapies (Table 1, Figure 2).

## 3. How to Guide Treatment Choice in First-Line Therapy

In recent years, some randomized phase 3 clinical trials have demonstrated the superiority of targeted agents compared to traditional immunochemotherapy in treatment-naive CLL patients. In patients aged 65 years or older, the RESONATE-2 study showed that single-agent ibrutinib is more effective than chlorambucil in terms of overall response rate (ORR), PFS and OS [54]. In the same setting, the ALLIANCE trial showed that ibrutinib, either alone or in combination with rituximab, confers a PFS advantage compared to bendamustine plus rituximab, without significant difference in terms of OS [55,56]. The final analysis of the iLLUMINATE trial showed as well that the combination ibrutinib–obinutuzumab was superior to chlorambucil–obinutuzumab in patients aged ≥65 years or <65 years with coexisting conditions, regarding PFS but not OS [57].

In younger patients, the results of the ECOG-ACRIN E1912 trial favored the combination ibrutinib–rituximab over chemoimmunotherapy (FCR) for both PFS and OS [58].

Next-generation BTKi has also shown a PFS advantage compared to immunochemotherapy, which has a reduced risk of adverse cardiovascular events. In particular, in the ELEVATE-TN trial, acalabrutinib with or without obinutuzumab significantly improved PFS over chlorambucil–obinutuzumab [59]. In the SEQUOIA trial, with a shorter follow-up, zanubrutinib significantly improved PFS over bendamustine–rituximab [60].

In the CLL14 trial, the combination venetoclax–obinutuzumab with a fixed duration of 12 months demonstrated a significantly longer PFS compared to chlorambucil–obinutuzumab in treatment-naive CLL patients with coexisting medical conditions [61]. A recent update of this study confirmed sustained and deep responses [62].

First-line venetoclax-based therapies have also been studied in fit patients [63]. The CLL13-GAIA trial randomly assigned fit patients with CLL, without *TP53* aberrations, to receive 6 cycles of chemoimmunotherapy (FCR if ≤65 years old or bendamustine–rituximab if > 65 years old) or 12 cycles of venetoclax–rituximab, venetoclax–obinutuzumab, or venetoclax–obinutuzumab–ibrutinib. Three-year PFS was significantly higher with venetoclax–obinutuzumab–ibrutinib (90.5%) and venetoclax–obinutuzumab (87.7%) compared to chemoimmunotherapy (75.5%) but with a higher rate of adverse events (particularly infections) in the triple therapy arm [64].

Based on the above trials, the treatment landscape of CLL has progressively abandoned in the last few years the traditional chemoimmunotherapy in favor of new biological agents and now basically includes fixed-duration therapies based on venetoclax plus anti-CD20 or continuous therapies based on BTKi, such as ibrutinib, acalabrutinib (with or without obinutuzumab), or zanubrutinib.

A third option, based on a time-limited therapy with 3 cycles of ibrutinib lead-in followed by 12 cycles of ibrutinib plus venetoclax, has recently been approved, thanks to the results of the GLOW and CAPTIVATE studies [65,66]. GLOW is a phase 3 trial that enrolled 211 patients older than 65 years or younger but with significant comorbidities, randomly assigned to ibrutinib–venetoclax or chlorambucil–obinutuzumab; cases with del(17p) were excluded, and few patients had *TP53* mutations. The all-oral ibrutinib–venetoclax combination demonstrated deeper and longer responses; 42-month PFS rates were 75% for ibrutinib–venetoclax and 25% for chlorambucil–obinutuzumab [65,67]. There were some concerns about cardiovascular adverse events, but they occurred in an elderly population with a high CIRS (Cumulative Illness Rating Scale) score and/or poor performance status [65]. CAPTIVATE is a phase 2 trial that enrolled fit patients aged ≤70 years and proposed the same treatment scheme as in the GLOW trial. The primary endpoint was met with a complete remission rate (CR) of 56% in patients without 17p deletions [66]. A recent update at the last ASCO meeting reported a 4-year PFS of 79% and a 4-year OS of 98% [68]. In both trials, the ibrutinib lead-in phase permitted a meaningful reduction of tumor burden and tumor lysis syndrome risk [65,66].

The choice between these three different treatment strategies should consider the patient’s preferences, comorbidities, fitness status, caregiver availability, and logistic and pharmacoeconomic considerations. Biological parameters also play a crucial role in the decisional therapeutic algorithm. The 2024 update of the ESMO Clinical Practice Guidelines recommends a pre-treatment evaluation of the patient’s biological risk profile, including the assessment of *IGVH* mutational status and *TP53* status (del(17p)—and/or *TP53* mutations) [69]. The ESMO treatment algorithm for first-line therapy considers, as decision-making points, the patient’s fitness status (fit or younger patients vs. unfit or older patients), the presence of *TP53* mutations or del(17p) and, in the absence of *TP53* variants, the *IGVH* mutational status (M vs. UM). With the exception of cases with TP53 mutations/del(17p), time-limited therapies should be preferred.

### 3.1. The Role of TP53 Mutations and/or del(17p) in the Choice of First-Line Therapy

The poor prognosis of CLL cases harboring *TP53* mutations and/or del(17p) has been known for many years [70]. Given the disappointing results of the traditional treatments, patients with *TP53* aberrations have often been excluded from randomized clinical trials testing new biological agents versus chemoimmunotherapy as a comparator arm. Therefore, recommendations for *TP53* mutated/deleted CLL patients are not based on large comparative series but on small studies, including limited numbers of such cases or sub-analysis of mixed cohorts.

Two phase 2 studies, including small cohorts of CLL patients with *TP53* alterations, reported good long-term outcomes with ibrutinib first-line, alone, or in combination with rituximab, with a 6-year PFS of 61% and 6-year OS of 79% [71,72]. A pooled analysis has recently included 89 CLL patients with *TP53* mutations and/or del(17p) receiving single-agent ibrutinib or ibrutinib in combination with anti-CD20 antibody across four studies: PCYC-1122e, RESONATE-2, iLLUMINATE and ECOG-ACRIN E1912. The PFS and OS estimates at 4 years were 79% and 88%, respectively [73]. Although early studies with BTKi in relapsed/refractory CLL (i.e., RESONATE trial) suggested inferior survival in the case of *TP53* aberrations [74], subsequent first-line studies showed similar outcomes. The final analysis of the iLLUMINATE study, stratified by *TP53* mutational status, demonstrated no significant differences in PFS between patients with and without *TP53* aberrations [57]. These results have been confirmed by the ALLIANCE study, where no differences in PFS were observed in patients treated with ibrutinib +/− rituximab based on *TP53* mutational status [55]. These data suggest that continuous treatment with BTKi may overcome the poor outcome pertaining to CLL cases with *TP53* disruptions.

The ELEVATE-TN trial included 73 treatment-naive CLL patients with *TP53* mutations and/or del(17p); this study did not compare patients with and without *TP53* aberrations but described a very good PFS (73 months) with acalabrutinib even in the presence of such unfavorable alterations [59]. The safety and efficacy of zanubrutinib in treatment-naïve patients with del(17p) were reported in a dedicated, nonrandomized cohort (arm C) of the phase 3 SEQUOIA trial. A total of 109 patients with centrally confirmed del(17p) were enrolled; the median PFS was not reached, and the estimated 18-month PFS rate was 88.6% [75]. 

CLL cases with *TP53* aberrations have a less favorable outcome with fixed-duration venetoclax-based therapies. In the previously described CLL-14 trial, patients with *TP53* mutations/del(17p) had a much better 5-year PFS when treated with venetoclax–obinutuzumab compared to chlorambucil–venetoclax (40.6% and 15.6%, respectively). However, the 5-year PFS achieved with venetoclax–obinutuzumab in patients with *TP53* mutations/del(17p), with the limits of indirect comparison, seems to be lower than that described in trials with continuous BTKi (about 70%) and significantly inferior compared to patients without *TP53* mutations/del(17p) treated with venetoclax–obinutuzumab in CLL-14 (65.8%) [62].

A subgroup analysis of high-risk patients in the fixed-duration cohort of CAPTIVATE trial showed a high rate of CR (52%) and uMRD (83% in peripheral blood, 45% in bone marrow) also in patients with *TP53* mutations and/or del(17p) [76]. However, a recent update at the last ASH meeting pointed out a numerically lower rate of 5-year PFS among these cases (45%) compared to the whole series (70%) [77].

Most data suggest that continuous treatment with BTKi should be the first choice for patients with *TP53* mutations and/or del(17p), as also recommended by ESMO guidelines [69]. Ibrutinib has the longest follow-up, but the known cardiovascular adverse events of this first-generation BTKi (e.g., atrial fibrillation up to 17%, hypertension up to 23%) make the use of second-generation, more selective BTKi (such as acalabrutinib or zanubrutinib) preferable, especially in elderly patients or those with comorbidities [54,55].

### 3.2. The Role of IGVH Mutational Status in the Choice of First-Line Therapy

CLL patients with M-*IGVH* and without *TP53* disruptions have a very favorable prognosis. The recently updated long-term follow-up results of the CLL8 study showed that the FCR regimen permits the achievement of durable remissions in M-*IGVH* cases, with an apparent plateau in PFS curves after 7 years and a possible cure for a significant part of these patients [78,79]. Results are similar or even better with biological agents. In the ECOG ACRIN E1912 trial, ibrutinib–rituximab led to superior PFS compared to FCR regardless of *IGVH* mutational status [80]. In the FLAIR trial, the PFS was significantly better with ibrutinib–rituximab compared to FCR in patients with UM-*IGVH* but not in M-*IGVH* cases [81]. Results for PFS also favored ibrutinib–venetoclax as compared with FCR in patients with UM-*IGHV* but not in those with M-*IGHV* [82]. However, besides this long-term efficacy of FCR, in particular in M-CLL, clinicians must consider a high rate of hematological toxicity, infections and second malignancies. For this reason, the ESMO guidelines still suggest FCR as a possible option for fit younger patients with a good biological profile, but only where targeted therapies are not reimbursed [69].

The CLL14 and CLL13-GAIA studies reported superior PFS with time-limited venetoclax plus anti-CD20 compared to chemoimmunotherapy, and results were particularly excellent for M-CLL [61]. In CLL14, the combination venetoclax–obinutuzumab demonstrated a better PFS vs. chlorambucil–obinutuzumab for both M-*IGVH* and UM-*IGVH* cases, but the curves showed a significant gap according to *IGVH* mutational status. The 5-year PFS for patients treated with venetoclax–obinutuzumab was 76.5% in M-*IGVH* vs. 55.8% in UM-*IGVH*; however, the 5-year PFS increases up to 59.4% if we consider UM-*IGVH* patients without *TP53* mutation/deletion and venetoclax–obinutuzumab remains a good option for these cases, considering the advantages of a time-limited chemo-free treatment [62].

BTKi can probably overcome the unfavorable predictive value of a UM-*IGVH* status. For example, survival curves with acalabrutinib ± obinutuzumab in the ELEVATE TN study were very similar in M-*IGVH* and UM-*IGVH* patients [59]. However, continuous treatment with BTKi is associated with significant costs, a higher risk of long-term toxicities, and an ongoing selection pressure that could impact clonal evolution. The economic and toxic burden of continuous BTKi therapy deserves particular attention in younger patients with long life expectancy.

Outcomes according to *IGHV* mutational status with the new combination ibrutinib–venetoclax in elderly CLL patients have been reported by the GLOW study, showing a better PFS in M-*IGVH* patients compared to UM-*IGVH* [67]. The CAPTIVATE trial in young CLL patients, similarly, showed higher uMRD rates in UM-*IGVH* vs. M-*IGHV* cases but a more rapid reappearance of leukemic cells over time; estimated 24-month PFS rates were 93% for UM-*IGVH* patients vs. 97% for M-*IGVH* patients [83]. The ESMO guidelines suggest the time-limited combination venetoclax–ibrutinib as a choice especially appropriate for fit/young CLL patients with UM-*IGVH* and no *TP53* mutation/deletion [69].

We are waiting for the important results of the CLL17 trial (NCT04608318), comparing the efficacy (primary endpoint PFS) of continuous ibrutinib monotherapy vs. fixed-duration venetoclax–obinutuzumab vs. fixed-duration ibrutinib–venetoclax.

## 4. How to Guide Treatment Choice in Relapsed/Refractory Disease

Relapsed CLL is defined as a disease recurrence after at least 6 months from a partial or complete remission. Instead, refractory disease represents non-response to therapy or progression within 6 months of completion of a time-limited therapy [84]. Progressive disease does not always require treatment but follows indications based on 2018 iwCLL criteria [13].

Before starting each new therapy, it is recommended to repeat genomic assessment, including TP53 mutation and karyotype/FISH, while *IGHV* mutational status does not change. In case of elevated serum Lactate Dehydrogenase (LDH), an asymmetrical rapid increase in lymph node size, B symptoms and/or hypercalcemia, it is also important to rule out a Richter’s transformation by Positron Emission Tomography (PET) scan and re-biopsy of sites with maximum Standardized Uptake Value (SUV max) ≥5. It is strongly advised to define the clonal relation between CLL and subsequent Diffuse Large B-cell lymphoma by comparing *IGHV* sequences; the prognosis is poorer in clonally related cases, in patients who previously received treatment for CLL and in the presence of *TP53* disruptions [85].

In general, also in the relapse setting, target therapy should always be preferred to chemoimmunotherapy, as the latter is inferior with respect to PFS and OS [86,87]. 

The main options in relapsed/refractory (r/r) CLL patients are venetoclax–rituximab and covalent BTKi (cBTKi). A time-limited therapy with ibrutinib–venetoclax is not yet approved in relapsed patients.

MURANO is a phase 3 trial comparing a time-limited chemo-free therapy with venetoclax–rituximab (venetoclax for up to 2 years and rituximab for the first 6 months) versus bendamustine–rituximab in r/r CLL patients [88]. The seven-year PFS rate was 23.0% with venetoclax–rituximab, while no cases treated with bendamustine–rituximab were progression-free at this time point. The study also showed a meaningful improvement in OS: 7-year OS rates were 69.6% with venetoclax–rituximab versus 51.0% with bendamustine–rituximab [89].

Since the approval of the first-in-class agent (ibrutinib) in 2014, BTKi has revolutionized the treatment of CLL, including the relapsed setting. In the RESONATE study [74], the superiority of ibrutinib compared to ofatumumab in r/r patients was maintained in high-risk molecular/cytogenetic groups (such as UM-*IGHV*, *TP53* mutation, del(17p), and del(11q)) with a median PFS of 44.1 vs. 8.0 months.

A more selective cBTKi, acalabrutinib, was compared with ibrutinib in patients with high-risk r/r CLL in the ELEVATE-RR (ACE-CL-006) phase 3 trial [90]. Treatment with acalabrutinib showed a non-inferior PFS and OS. However, acalabrutinib had a better safety profile compared with ibrutinib since a lower discontinuation rate due to adverse events was described (14.7 versus 21.3%). Based on these data, acalabrutinib represents a suitable alternative to ibrutinib, especially in patients with cardiologic comorbidities or who are intolerant to the first-in-class BTKi. This was confirmed in the phase 2 study ACE-CL-208, which evaluated acalabrutinib in patients with r/r CLL and intolerance to ibrutinib, showing an ORR of 73% [91].

Zanubrutinib, a second-generation cBTKi, was compared with ibrutinib in the phase 3 ALPINE trial in the setting of r/r CLL, demonstrating superiority in terms of efficacy and safety [92]. Zanubrutinib was better tolerated compared to ibrutinib in terms of adverse events leading to discontinuation (16.2 vs. 22.8%). Atrial fibrillation/flutter was also significantly lower with zanubrutinib versus ibrutinib (5.2 vs. 13.3%). ORR and PFS, at 24 months, were significantly higher with zanubrutinib versus ibrutinib across all risk groups. These results led to regulatory approval of zanubrutinib in r/r CLL patients.

While the choice of first-line therapy is more codified based on biological and clinical parameters, second-line therapy is less standardized and should be chosen taking into consideration many factors, mainly related to the type of previous lines of therapy, quality and duration of response, tolerance, access to targeted agents and availability of clinical trials. The recently updated ESMO Clinical Practice Guidelines stratify patients at relapse into four categories: relapse after chemoimmunotherapy or late (≥36 months) relapse after venetoclax-based time limited therapy without *TP53* mutation or del(17p); early relapse (< 36 months) after venetoclax-based time-limited therapy; relapse or progression while on BTKi treatment; relapse or progression with *TP53* mutation or del(17p) [69].

### 4.1. The Role of TP53 Mutations and/or del(17p) in the Choice of Second-Line Therapy

Relapsed patients with *TP53* mutation and/or del(17p), which are cBTKi anti-BCL-2 naïve, regardless of *IGHV* mutational status, should be treated with acalabrutinib, zanubrutinib or ibrutinib; alternatively, with venetoclax–rituximab. Venetoclax in monotherapy or idelalisib–rituximab may be a second option [69].

BTKi has been evaluated in high-risk CLL patients harboring TP53 disruptions. In the final analysis of the RESONATE trial, with 6 years of follow-up, no significant difference in PFS for ibrutinib-treated patients with or without del(17p) was confirmed [86]. In the open-label, phase 2, RESONATE-17 study, ibrutinib demonstrated in del(17p) r/r CLL patients an ORR of 83% and 24-month PFS and OS rates of 63% and 75%, respectively [93]. In the ALPINE study, zanubrutinib significantly improved PFS and ORR compared with ibrutinib and this benefit was observed across all subgroups, even among patients with del(17p)/*TP53* mutation [92]. 

The ASCEND study showed the superior efficacy of acalabrutinib compared to idelalisib–rituximab or bendamustine–rituximab in relapsed/refractory patients, including those with high-risk disease (complex karyotype, UM-*IGHV*, *TP53* mutation and/or del(17p)). The final analysis of this study, with a follow-up of 4 years, confirmed the advantage of acalabrutinib in all subgroups [94]. In particular, median PFS was 45.5 months vs. 11.1 months in the subgroup with del17 and/or *TP53* mutation.

In the MURANO trial, the presence of high-risk cytogenetic and/or molecular abnormalities was associated with a significantly shorter median PFS among patients treated with fixed-duration venetoclax–rituximab; in particular, in patients with del(17p) and/or *TP53* mutation, the median PFS was 37.4 months versus 56.6 months in *TP53*-wt patients [87].

Venetoclax monotherapy was explored in r/r CLL patients harboring del(17p) by Stilgenbauer et al., who reported 6-year follow-up data and subgroup analyses of a phase 2 trial (M13-982) in which a total of 153 patients received 400 mg venetoclax monotherapy until progressive disease [95]. The best ORR was 77%, with a median duration of response of 39.3 months. With the limits of a cross-trial comparison, outcomes with venetoclax monotherapy in this study appear comparable with ibrutinib in patients with r/r CLL and del(17p)/TP53 mutation of RESONATE-17 trial and with zanubrutinib efficacy data in ALPINE trial. Further studies are still warranted to define the optimal treatment for these patients.

### 4.2. The Role of Prior Lines of Therapy

In patients without *TP53* alterations and previously treated only with chemoimmunotherapy, the choice of a second-line treatment in case of relapse or progression is like that described for the first-line setting. This scenario is expected to be very limited in the coming years because targeted agents are now the standard first-line treatment. Important aspects such as treatment duration, way of administration, frequency of clinical and biochemical controls and side effects should always be discussed with the patient and caregivers.

If the previous treatment was venetoclax-based, a switch to BTKi could be recommended (acalabrutinib and zanubrutinib preferred over ibrutinib). A re-treatment with venetoclax may be considered if the prior anti-BCL2 therapy was fixed-duration and the duration of remission was >24–36 months [69].

If the previous treatment was a continuous cBTKi-based therapy, it is essential to discriminate between interruption due to progression or intolerance/toxicity. In the first case, the preferred choice is a venetoclax-based therapy, although the efficacy is lower in BTKi refractory patients. In the second case, a switch to alternative cBTKi is a feasible choice [69].

Patients progressing after BTKi and anti-BCL2 are a difficult-to-treat population. The use of allogeneic HSCT can be suggested for younger and fit patients (<70 years) with prior exposure to ≥2 targeted therapies, especially in case of short response and/or high-risk biologic disease, such as *TP53* mutations/del(17p) [69]. It remains a possible therapy option as an alternative to CAR-T cells, which was recently approved by the Food and Drug Administration (FDA) [96,97].

### 4.3. Mechanisms of Resistance to Target Therapy and New Drugs

Despite the high efficacy of target therapy, primary and acquired resistance has been described.

Data from prospective clinical trials report that more than 50% of patients discontinue ibrutinib within 5 years of treatment [56,98]. The major reasons for discontinuation were disease progression and adverse events (particularly atrial fibrillation, infections, hemorrhage, and diarrhea).

Complex karyotype, del(17p) and *BCL6* defects were identified as poor prognostic factors for the development of secondary resistance. Moreover, an increasing ratio of mutations in *SF3B1*, *MGA*, *BIRC3*, *NFKBIE*, *CARD11*, and *XPO1* was associated with acquired ibrutinib resistance [99]. 

In case of progression during BTKi therapy, before starting an alternative covalent BTKi, it is recommended to test for acquired resistance mutations. These mutations are not present before treatment but can be found in 57–87% of patients progressing on ibrutinib and at a similar frequency on acalabrutinib, implying treatment resistance for the entire class of currently covalent BTKi [100,101,102]. C481S/R/F/Y is the most common hotspot found in the *BTK* gene, located in the binding site for ibrutinib, causing a reduction of ibrutinib affinity. Mutations outside of the kinase domain are very rare. While *BTK* C481 mutations are shared by both cBTKi, alternative uncommon BTK variants are described as associated with acalabrutinib (T474I and E41V) or ibrutinib (L528W, A428D) [103].

A second resistance mechanism consists of the acquisition of *PLCG2* mutations in CLL patients who failed on ibrutinib treatment. *PLCG2* encodes for a phospholipase Cγ2 protein, which is immediately downstream of *BTK*. *PLCG2* hotspot mutations have a gain-of-function effect, resulting in continuous BCR signaling. *BTK* and *PLCG2* mutations are usually acquired between the second and fourth year on ibrutinib, with a higher frequency in r/r patients and in high-risk cytogenetics categories. However, the clinical impact of *PLCG2* mutations remains not clearly understood [101,104,105].

Although mutations in *BTK* and *PLCG2* genes constitute up to 80% of all resistances, for the remaining 20%, the mechanism of ibrutinib resistance is still unknown [106].

Other mechanisms of resistance may involve targets of venetoclax. The most common one includes a G101V-acquired hotspot mutation in the *BCL2* gene, which results in impaired interaction between venetoclax and its binding site. Alternative mechanisms of resistance, such as mutually exclusive or coexisting non-G101V *BCL2* mutations, upregulation of other anti-apoptotic proteins (e.g., MCL-1 and BCL-xL), and reprogramming of mitochondrial oxidative processes have been described [107].

The use of fixed-duration venetoclax-based combinations may reduce the likelihood of developing new CLL-targeting mutations, thus improving clinical responses upon retreatment with venetoclax. The acquisition of the recurrent G101V mutation in the *BCL2* gene has been reported only with continuous venetoclax and never following 12 to 24 months of fixed-duration therapy [108].

Considering that resistance-associated mutations often occur at low VAF, it is necessary to apply highly sensitive approaches, such as the NGS technique. *BTK*, *PLCG2* and *BCL2* mutations can be predictive of relapse, even if they occur several months before clinical symptoms. Additionally, understanding the mechanisms behind resistance to these novel agents could allow clinicians to find new strategies to prevent resistance before it develops or to manage it better in case of onset [104,105].

One way of overcoming resistance mediated by *BTK* mutations is the use of reversible, non-covalent BTKi developed not to engage on the C481 binding site, and therefore, effective on both C481-mutant and unmutated *BTK* with similar efficacy.

Two agents, pirtobrutinib (LOXO-305) and nemtabrutinib (ARQ-531) demonstrated safety and efficacy in early-phase clinical trials, including CLL cases with C481 mutated BTK [109,110].

Pirtobrutinib showed an ORR of 62% in the phase I/II BRUIN study in CLL patients previously exposed to BTK inhibitors [110]. Response to pirtobrutinib was not impaired by previous resistance to cBTK inhibitors.

Nevertheless, acquired resistance to pirtobrutinib has also emerged. Several non-C481 mutations have been described in patients who progress during pirtobrutinib therapy. L528W was commonly found in patients resistant to cBTKi (ibrutinib and zanubrutinib) and non-covalent BTKi (pirtobrutinib); the position of *BTK* mutations may vary for different BTK inhibitors [111].

NX-2127 is a small molecule characterized by a novel mechanism of BTK degradation and is currently investigated in a phase 1 study (NCT04830137) [112]. A first analysis reported an ORR of 33%, and responses were registered in BTKi/BCL2 double refractory patients and those who progressed on ncBTKi.

Another way to overcome resistance to target therapy is a combination of different molecules, with the goal of achieving MRD eradication to allow treatment discontinuation and prevent acquired mutations.

Different treatment combinations are objects of study. The phase 2 CLARITY study evaluated the fixed-duration combination of ibrutinib and venetoclax in r/r CLL, with a good safety profile and deep MRD eradication [113]. This all-oral, fixed-duration combination is not presently approved in relapse settings but only in front-line patients based on the results of the GLOW and CAPTIVATE studies [66,67]. A triplet combination of ibrutinib, venetoclax, and obinutuzumab resulted in an ORR of 88% in r/r CLL patients, with no relevant toxicity, except for myelosuppression [114].

## 5. The Emerging Role of MRD: Are We Ready for Clinical Use?

MRD is defined as the presence of residual cancer cells after treatment in patients with clinically undetectable disease. According to the proportion of residual detectable CLL cells, MRD is defined as follows: MRD4 is one leukemic cell in 10,000 leukocytes (1 in 10^−4^ or 0.01%), MRD5 is one leukemic cell in 100,000 leukocytes (1 in 10−5 or 0.001%), and so on. MRD4 is the current cut-off for undetectable MRD (uMRD) and has the potential to become a reproducible prognostic stratification factor for PFS after treatment. However, multiple aspects and potential limits are still under evaluation, such as sample volume, type of tissue (bone marrow or peripheral blood), assay time, and use of cellular (flow cytometry) versus molecular (ASO- PCR, NGS) tests [115,116].

MRD data in CLL were first derived from the immunochemotherapy era when an MRD evaluation (for example, after FC or FCR treatment in the CLL8 trial) was proved to be a reliable, independent prognostic marker of PFS and OS [117]. 

Although current guidelines do not recommend routine MRD testing in clinical practice, it has been included in recent clinical trials as a primary or secondary endpoint (Table 2).

In the phase 3 MURANO trial, the primary endpoint was investigator-assessed PFS, but MRD evaluation was a key secondary endpoint, although not determinant for treatment duration. Seymour et al. reported a detailed analysis of MRD kinetics in the MURANO study with a 5-year follow-up, including a subgroup analysis of patients with high-risk biomarkers [87]. The venetoclax–rituximab MRD level was confirmed to be a robust predictor of PFS and OS. Patients who achieved an uMRD at the end of therapy (EOT) had superior PFS and OS, suggesting the importance of MRD eradication as a goal of treatment. Particularly durable PFS was observed among patients with M-*IGHV* who completed treatment and had uMRD at EOT. At the 5-year update, 32 patients (38.6%) continued to show uMRD without clinical progression, while 47 patients (56.6%) had MRD conversion. The median time from EOT to MRD conversion was 19.4 months, and the median time to clinical progression was even longer (25.2 months).

These data make it evident that MRD can be a powerful prognostic tool to predict outcomes and is now considered a surrogate marker to assess treatment efficacy in randomized trials before clinical endpoints.

Several are the potential clinical implications of MRD in the near future: MRD could be useful in deciding when to stop treatment; the longitudinal monitoring of MRD will also allow the study of the kinetics of the disease and could guide the rechallenge of therapy after treatment discontinuation. Furthermore, the MRD level combined with screening mutations could guide the switching to a non-cross-resistant agent or combinations of agents to anticipate the clinical relapse.

One of the main examples of an MRD-guided therapeutic strategy is the FLAIR trial, where ibrutinib–venetoclax was compared to FCR and the duration of ibrutinib–venetoclax therapy was defined by MRD evaluation. The primary endpoint was PFS in the ibrutinib–venetoclax group as compared with the FCR group. MRD-directed ibrutinib–venetoclax therapy has been demonstrated to improve PFS and OS if compared with FCR [81].

In a follow-up analysis of the randomized GLOW study, a strong interaction between *IGHV* status and end-of-treatment MRD status was found [67]. UM-*IGHV* patients had uMRD levels of 60%, while M-*IGHV* patients had uMRD levels of 40%. Despite this, the PFS of UM-*IGHV* patients was shorter than M-*IGHV* ones, suggesting that in the context of BCL2i + BTKi, the prognostic utility of MRD is dependent on the *IGHV* status because it determines the kinetics of MRD negativity and disease progression. This is in line with the previous observation of venetoclax–obinutuzumab showing MRD doubling time strongly associated with the presence of high-risk genetic characteristics such as *IGHV* mutational status or complex karyotype [57,118,119].

## 6. Conclusions and Future Directions

In recent years, there has been rapid progress in new molecular methods to analyze the genetic landscape of CLL [41]. As a result, new prognostic/predictive biomarkers and new targeted therapies have been developed, resulting in improved risk stratification tools, prolonged PFS and extended OS in CLL patients with a chemo-free approach. Considering the new technologies available to date, there is a clear need for a more complete biomarkers assessment at diagnosis. Despite the high efficacy of current treatments, CLL remains, in most cases, an incurable disease. Molecular studies are also crucial to decipher mechanisms of clonal evolution and therapeutic resistance [120].

To date, an inherited susceptibility to CLL has been described in a few studies, where a possible role for germline variants in the *ATM* and *XPO1* gene is suggested. This aspect is still challenging because a germline origin needs to be confirmed on an alternative tissue and because these patients may require specific clinical management [42,45].

Although an MRD assessment is not yet recommended for clinical decisions, it may acquire a key role in the next few years to establish patient responses and prognoses. While flow cytometry is now considered the gold standard for MRD testing, the improvement of more sensitive technologies such as NGS will allow a higher and deeper detection of residual leukemic cells for more precise patient management [115,121,122].

## Figures and Tables

**Figure 1 cancers-16-03483-f001:**
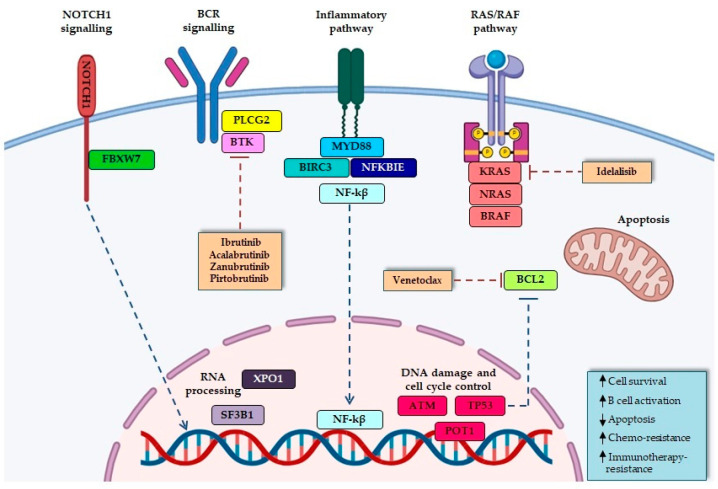
Main pathways involved in CLL. The illustrated genes are prognostic markers and some of them therapeutic targets.

**Figure 2 cancers-16-03483-f002:**
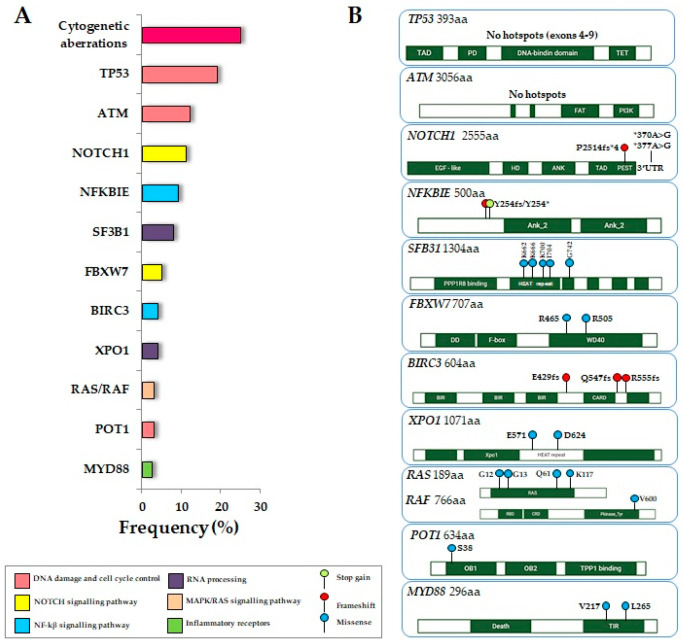
Driver genes and recurrent somatic variants in CLL. (**A**): Representation of the frequency of the main genetic alterations in CLL. (**B**): Boxes summarise the key protein domains and variants most frequently observed in the listed genes.

**Table 1 cancers-16-03483-t001:** Clinical significance in CLL prognosis of the different genetic and molecular markers.

Biomarker	Detection Method *	Clinical Significance in Prognosis
Unmutated *IGHV*	Sanger, NGS	Shorter TTFT and poorer response to CIT Mandatory in pre-treatment evaluation (stable status during disease course)
Del(17p)/*TP53* mutation	FISH + Sanger, NGS	Resistance to CIT and rapid disease progression Mandatory in pre-treatment evaluation
Complex Karyotype	Conventional Karyotyping	Unfavorable outcome after CIT, independently of TP53 alterations; Controversial role after novel targeted agents
*NOTCH1* mutation	Sanger, NGS	Worse outcome and poor response to rituximab treatment
*FBXW7* mutation	Sanger, NGS	Poorer PFS and OS in patients with early-stage disease
*SF3B1* mutation	Sanger, NGS	Poor prognosis
*XPO1* mutation	Sanger, NGS	Inferior PFS and OS. Independent prognostic variables
Del(11q)/*ATM* mutation	FISH + Sanger, NGS	Shorter TTFT but better response to BTK inhibitors in the presence of del(11q) Germline mutations could be detected
*POT1* mutation	Sanger, NGS	Poor OS Germline mutations could be detected
*BIRC3* mutation	Sanger, NGS	Unfavorable prognosis, but not confirmed across literature; probable predictive role
*NFKBIE* mutation	Sanger, NGS	Reduced response to ibrutinib treatment
*MYD88* mutation	Sanger, NGS	Good prognosis
*RAS/RAF* mutation	Sanger, NGS	*BRAF*: adverse OS *NRAS/KRAS*: no adverse OS

* Limit of detection (LOD) of the cited techniques: Sanger sequencing: LOD 10–15%; NGS sequencing: LOD 0.1% with adequate bioinformatic pipeline, limits in big indel mutations detection; FISH: LOD 100 kb–1 Mb of DNA; Conventional karyotyping: LOD more than 5 Mb of DNA.

**Table 2 cancers-16-03483-t002:** List of the main clinical trials in which MRD assessment is included as an endpoint and the respective molecular methods used for evaluation.

Trial	Arms	Phase	Total Participants	MRD As a Primary Outcome	MRD as a Secondary Outcome	MRD Stopping Rules	MRD Assessment
MURANO	BR vs. VR	3	389	No	YES	No	FC, ASO
CLARITY	IV single arm	2	54	YES	No	YES	FC
CLL14	ClbO vs. VO	3	432	No	YES	No	FC, ASO, NGS
CAPTIVATE MRD	IV then I or placebo if U-MRD4	2	54	No	YES for the IV vs. placebo	No	FC
CAPTIVATE FD	IV	2	159	No	YES	No	FC
GLOW	IV vs. ClbO	3	211	No	YES	No	NGS, FC
FLAIR	FCR, IR, I, IV	3	771	YES for IV vs. 1 or 1R	YES for FCR vs. IR	YES for the I arms	FC
CLL13	CIT vs. VR vs. VO vs. VIO	3	926	YES	No	No	FC
GALACTIC	O consolidation	2	48	YES	No	No	FC

CIT—chemoimmunotherapy; Clb—chlorambucil; FCR—fludarabine, cyclophosphamide, and rituximab; I—ibrutinib; O—obinutuzumab; R—rituximab; V—venetoclax. FC—flow cytometry; ASO—Allele-Specific Oligonucleotides; NGS—Next-Generation Sequencing.

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
