# Peer review of "Recent Advances in the Molecular Biology of Chronic Lymphocytic Leukemia: How to Define Prognosis and Guide Treatment"

_cancers, 2024, doi:10.3390/cancers16203483_

Round 1
Reviewer 1 Report
Comments and Suggestions for Authors
In this review, the Authors very thoroughly described the current status of CLL biology, treatment of CLL patients and the prognostic markers. The genetic alterations associated with CLL were sufficiently discussed in light of current knowledge. In general, this is a well written manuscript.
I have only minor concerns:
1. Gene names should be written in italics.
2. The review is very detailed. In this respect, I would recommend to illustrate appropriate parts of the manuscript with relevant figures e.g., related to signaling pathways altered in CLL.
3. The Authors did not discuss certain potential markers e.g., considered in the past, including lipoprotein lipase. This issue might be refered to even if it is not currently important. In this type of review paper, which is designed to present the current state of knowledge, it is justified to discuss the current status of previously recognized markers.
Author Response
Comments 1: Gene names should be written in italics
Response 1: Thank you for your suggestion. We have written all gene names in italics
Comments 2: The review is very detailed. In this respect, I would recommend to illustrate appropriate parts of the manuscript with relevant figures e.g., related to signaling pathways altered in CLL.
Response 2: Thank you for this comment. We added two Figures to better illustrate some molecular sections of the manuscript. Figure 1 illustrates the main molecular pathways involved in CLL with prognostic significance and/or used as therapeutic targets. Figure 2 represents the main genetic alterations in CLL with the frequency of known variants.
Comments 3: The Authors did not discuss certain potential markers e.g., considered in the past, including lipoprotein lipase. This issue might be refered to even if it is not currently important. In this type of review paper, which is designed to present the current state of knowledge, it is justified to discuss the current status of previously recognized markers.
Response 3: Thank you for your comment about the role of other potential markers considered in the past. For lenght reasons, unfortunately, we had to consider only molecular biomarkers currently used in clinical practice or with a possible role in the next few years. Lipoprotein lipase is a central enzyme in lipid metabolism and its role in CLL has been discussed in previous studies. However, most papers described lipoprotein lipase expression, associated with an aggressive clinical course of CLL, not specific molecular alterations which are the focus of our review.
Reviewer 2 Report
Comments and Suggestions for Authors
Chronic Lymphocytic Leukemia (CLL) is the most common type of leukemia accounting for approximately 1% of all cancer cases. Quite frequently, before onset, CLL is preceded by a pre-neoplastic condition, namely monoclonal B cell Lymphocytosis. From the clinical point of view, CLL displays a quite heterogeneous behaviour and this is associated with the heterogeneous genomic aberrant landscape that is featuring CLL cells. Remarkably, in the last years we have witnessed a significant evolution of the the therapeutic approaches, alongside our deeper knowledge about broad spectrum of genomic alterations.
In the review paper titled "Recent Advances in the Molecular Biology of Chronic Lymphocytic Leukemia: How to Define Prognosis and to Guide Treatment" Arcari A. and colleagues first discuss some recent insights concerning the CLL biology and then they "consider the translation of these findings into the development of risk-adapted and targeted therapeutic approaches".
Overall, the novelty offered by the manuscript is quite poor, since the same issue has been extensively covered quite recently (e.g., Shadman M. in JAMA "Diagnosis and Treatment of Chronic Lymphocytic Leukemia", doi:10.1001/jama.2023.1946).
However, before manuscript publication, some issues require to be sorted out.
- First of all, the main text would greatly benefit from a significant "trimming" of at least 30-40%. By being a review manuscript it should be focused on a subject, it should be critical, and comparative. A detailed description is not required in a review article, because for details the readers will refer to the corresponding original paper listed in the References section. All details regarding all the percentages and the p-values render the manuscript too verbose and, eventually, one gets lost. A review should convey sharp and clear messages. Additionally, to some extent, some parts look like "frills" (e.g., lines 129-132).
- When used the first time acronyms must be spelled out (e.g., line 190: BCR; line 238: TTFT; line 281: NES; line 428: CIRS; Table 1: BE, and so on).
- Quite a few typos are scattered throughout the main text (e.g., lines 426, 801-802, etc…).
- Based on what is stated from line 266 to line 272 it is not clear why SF3B1 is classified as a component of ribosomal processing. Furthermore, always in the same paragraph, the authors should pay attention to not convey misleading messages. When the authors state that "The precise mechanism in which SF3B1 mutant alters its function is not yet clearly understood, but it seems that these mutations cause the disruption of the binding of the protein to some cofactors, affecting the structure and coding potential of gene transcripts across multiple pathways,…" they should be aware that SF3B1 mechanism of action has been clarified. The mutated forms of SF3B1 alter splicing mechanisms by recognizing a cryptic splicing site at the branch point of the intron.
- Lines 110-112: the authors are asked to clarify the sentence because as such sounds quite ambiguous.
The English language requires some amendments. A couple of examples are given. In the section Abstract (line 21) "…p53 disruptions…". According to what the authors then state in line 99 it may be that the words "mutations" or "deletions" fit better in the context. Line 219-220: what do the authors mean by "…genetic pathways…"? Cellular processes? Signaling pathways? Please check carefully throughout the main text.
Author Response
Comments 1: the main text would greatly benefit from a significant "trimming" of at least 30-40%. By being a review manuscript it should be focused on a subject, it should be critical, and comparative. A detailed description is not required in a review article, because for details the readers will refer to the corresponding original paper listed in the References section. All details regarding all the percentages and the p-values render the manuscript too verbose and, eventually, one gets lost. A review should convey sharp and clear messages. Additionally, to some extent, some parts look like "frills" (e.g., lines 129-132).
Response 1: Thank you so much for pointing this out. We agree with your suggestions. Accordingly, we have shortened many parts of the manuscript, trying to get it more clear and focused on the topic. We have deleted most of the percentages/HR/p values. We have also modified several sentences to make them more concise and easy for the readers. We have also deleted some lists of gene variants, moving them into a specific figure (Figure 2).
Comments 2: When used the first time acronyms must be spelled out (e.g., line 190: BCR; line 238: TTFT; line 281: NES; line 428: CIRS; Table 1: BE, and so on). Quite a few typos are scattered throughout the main text (e.g., lines 426, 801-802, etc…).
Response 2: We have re-checked all the acronyms, spelling them out when they first appear in the text. We have also corrected all typos.
Comments 3: Based on what is stated from line 266 to line 272 it is not clear why SF3B1 is classified as a component of ribosomal processing. Furthermore, always in the same paragraph, the authors should pay attention to not convey misleading messages. When the authors state that "The precise mechanism in which SF3B1 mutant alters its function is not yet clearly understood, but it seems that these mutations cause the disruption of the binding of the protein to some cofactors, affecting the structure and coding potential of gene transcripts across multiple pathways,…" they should be aware that SF3B1 mechanism of action has been clarified. The mutated forms of SF3B1 alter splicing mechanisms by recognizing a cryptic splicing site at the branch point of the intron.
Response 3: Thank you for your comment. Accordingly, we have changed the title of the 2.3.2 paragraph in "RNA processing". We have also significantly modified the SF3B1 paragraph, with a more correct definition of the mechanism of action: "SF3B1 represents the largest subunit of the SF3B complex and functions as a core component of the U2 snRNP, crucial for the branch site recognition and for the first stages of assembly of the spliceosome. It is recurrently mutated in approximately 8% of CLL patients. SF3B1 mutations mainly occur in the C-terminal HEAT-repeat domain (i.e. K700E); mutant SF3B1 seems to inhibit the canonical recognition by wild-type U1 snRNA on specific 5′ splice sites."
Comments 4: Lines 110-112: the authors are asked to clarify the sentence because as such sounds quite ambiguous.
Response 4: We agree. We have modified this sentence.
Comments 5: The English language requires some amendments. A couple of examples are given. In the section Abstract (line 21) "…p53 disruptions…". According to what the authors then state in line 99 it may be that the words "mutations" or "deletions" fit better in the context. Line 219-220: what do the authors mean by "…genetic pathways…"? Cellular processes? Signaling pathways? Please check carefully throughout the main text.
Response 5: Thank you for your suggestions. We replaced "TP53 disruptions" with "mutations" in the Abstract and modified other terms throughout the main text as suggested. The manuscript has been checked by an english native speaker.
Round 2
Reviewer 2 Report
Comments and Suggestions for Authors
Though the revised version of the manuscript titled "Recent Advances in the Molecular Biology of Chronic Lymphocytic Leukemia: How to Define Prognosis and to Guide Treatment", has been implemented with a couple of figures, and I appreciate them, I did not notice substantial "trimming" aimed to provide a sharper cut to the manuscript. The manuscript at the very best has been shortened by approximately less than 10%. It remains quite verbose.
Several typos are still scattered throughout the main text.
Author Response
Thank you for your comment. Following your suggestions, the last version of our manuscript has been significantly shortened (a further approximately 15%, in addition to the previous 10%). We hope that this new version will be appreciated as more concise and effective for the readers.